# Reinforcement Replaces Supervision: Query focused Summarization using Deep Reinforcement Learning

**Swaroop Nath♣, Pushpak Bhattacharyya♣, Harshad Khadilkar◇**
♣Computer Science and Engineering, IIT Bombay, India
◇TCS Research, India
{swaroopnath, pb}@cse.iitb.ac.in
harshad.khadilkar@tcs.com

## Abstract

Query-focused Summarization (QfS) deals with systems that generate summaries from document(s) based on a query. Motivated by the insight that Reinforcement Learning (RL) provides a generalization to Supervised Learning (SL) for Natural Language Generation, and thereby performs better (empirically) than SL, we use an RL-based approach for this task of QfS. Additionally, we also resolve the conflict of employing RL in Transformers with Teacher Forcing. We develop multiple *Policy Gradient networks*, trained on various reward signals: ROUGE, BLEU, and Semantic Similarity, which lead to a *10-point improvement* over the State-of-the-Art approach on the ROUGE-L metric for a benchmark dataset (ELI5). We also show performance of our approach in zero-shot setting for another benchmark dataset (Debate-Pedia) – our approach leads to results comparable to baselines, which were specifically trained on DebatePedia. To aid the RL training, we propose a better semantic similarity reward, enabled by a *novel Passage Embedding* scheme developed using Cluster Hypothesis. Lastly, we contribute a *gold-standard test dataset* to further research in QfS and Long-form Question Answering (LfQA).

## 1 Introduction

Query-focused Summarization (**QfS**) (Tombros and Sanderson, 1998; Dang, 2005; Nema et al., 2017) advances text summarization by letting the user provide a query, pertaining to which the summary must be generated from the document(s). Specifically, we target QfS for questions on a single document (see Table 1 for example). Our work is related to Long-form Question Answering (**LfQA**) (Fan et al., 2019; Krishna et al., 2021; Su et al., 2022). However, it differs from LfQA research significantly in that LfQA research has to focus on the retrieval of relevant passage(s) also. Whereas our system already assumes the presence of a document, which has to be summarized.

| |
|---|
| *Query*: *What is String Pool in Java?* |
| *Document*[1]: *String pool is nothing but a storage area in Java heap where string literals stores. It is also known as String Intern Pool or String Constant Pool. It is just like object allocation. By default, it is empty and privately maintained by the Java* ⋯ |
| *Summary*: *String Pool, used to reduce the memory footprint, is a specific area in the memory allocated to the process used to store String literal declared within Java program.* |

**Table 1:** Example for Query-focused Summarization for a well-formed question. The document is shortened (marked by ⋯) for space.

**Problem Statement:** We develop a system which abstractively generates summaries (Nallapati et al., 2016; Rush et al., 2015; Xu and Lapata, 2021, 2022) from a single document given the query. Specifically, **input**: *query* and *document*, and **output**: *summary*. We design a novel Reinforcement Learning algorithm for training and propose a novel passage embedding-based reward function.

**Motivation:** QfS lets the user drive the summarization. This improves the user experience by letting the user extract the needed information quickly. Additionally, present-day QA systems (Calijorne Soares and Parreiras, 2020) produce only short answers to factual questions. What they lack is the ability to consume a large piece of information and present a coherent, compressed summary based on the needs of the user. Our work aims to further research in LfQA by successfully building a QfS system for questions.

We utilize Reinforcement Learning (RL) for training the model, as RL provides a generalized loss to train our model. While Cross-Entropy loss

---

[1]Source: `javatpoint.com/string-pool-in-java`

utilizes token-by-token match, RL helps us generalize this further to phrase/sentence/semantic match. Such a generalization gives the model more freedom on what to generate at each time step, as long as the final generation suffices. Inspired by this, we employ RL for QfS. We detail on how RL is a generalization to Supervised Learning (SL) using Cross-Entropy loss in Section 3.1.

However, employing RL for text generation using Transformers is non-trivial. RL helps train agents that perform a sequence of actions, each deciding the next state and the available set of actions. In contemporary text generation through Transformers, Teacher Forcing[2] (Williams and Zipser, 1989) is used, where the generation (action) at time-step $t-1$ has no influence on the generation at time-step $t$. Effective utilization of RL for text generation needs to model this influence, thus necessitating the omission of Teacher Forcing. However, this increases the training time and memory footprint significantly. We propose a way to employ RL for text generation without Teacher Forcing, based on Scheduled Sampling (Bengio et al., 2015), Section 3.1. To the best of our knowledge, **we are the first work to resolve this conflict of employing RL in Transformers with Teacher Forcing**.

Our contributions are:

1. An RL algorithm that resolves the conflict of RL-based training in Transformers with Teacher Forcing. We observe significant improvement over the State-of-the-Art SL models: **37.2**% improvement in automatic evaluations (Table 6), **19**% improvement in Correctness (human evaluation; Table 8).

2. A human-curated test set, **250 instances**, devoid of Topic Centralization (one document can cater to multiple queries, Baumel et al. (2016)), for analysis of QfS models.

3. A passage embedding based novel reward mechanism (2.21 ROUGE-L improvement over ROUGE reward; Table 6) to score generated summaries, trained using the long-standing, time-honored Cluster Hypothesis (Jardine and van Rijsbergen, 1971).

4. A new dataset, $\sim$ **8 million instances**, to train a passage embedder model, scraped from `reddit`.

---

[2]Teacher Forcing refers to the way of training Autoregressive models where generation of $y_t$ (current token) depends on $y_1, y_2, \cdots, y_{t-1}$ (true previous tokens), and not on $\hat{y}_1, \hat{y}_2, \cdots, \hat{y}_{t-1}$ (generated previous tokens).

## 2 Related Works

**Abstractive QfS:** Supervised Learning approaches to abstractive QfS have been primarily limited by the availability of large-scale datasets. Dang (2005) present the first QfS dataset with open-ended questions. However, the size of the dataset is insufficient to train neural models. Xu and Lapata (2021, 2022); Laskar et al. (2020) tackle the scarcity of data through innovative approaches. Xu and Lapata (2021) generate proxy queries from generic summarization datasets by using a masked representation to train a model for QfS. Xu and Lapata (2022) model the document as an indicator of all the possible queries on it, and develop a QfS model by modeling latent queries through variational inference techniques. Laskar et al. (2020) take the approach of Transfer Learning: they fine-tune an abstractive summarizer, trained on XSum (Narayan et al., 2018), on a decent-sized (12695 instances) dataset: DebatePedia (Nema et al., 2017). Inspite of the presence of DebatePedia, researchers have mainly focused on leveraging generic summarization datasets due to the size limitation. We circumvent this limitation by utilizing ELI5 (Fan et al., 2019). Specifically, we utilize the version by Fan et al. (2019), not the KILT version (Petroni et al., 2021), as the latter does not provide gold documents to generate the summary from. Although, it is a dataset for LfQA, the format of the dataset makes it suitable for our use case: QfS for question on a single document.

ELI5 is an automatically curated dataset with a few shortcomings, which make it unsuitable for testing models (Krishna et al., 2021). Motivated by these shortcomings, we propose a novel, human-curated test set for QfS on well-formed questions consisting of high-quality 250 instances.

**RL for Summarization/QfS:** The usage of RL in QfS has been limited to extractive summarization only (Mollá and Jones, 2019; Mollá et al., 2020; Chali and Mahmud, 2021; Shapira et al., 2022). Mollá and Jones (2019); Mollá et al. (2020); Shapira et al. (2022) use RL to train sentence selector models, which select sentences to be incorporated into the summary. Chali and Mahmud (2021) present a hybrid summarization, where an extractive module selects text from the document, which is then used by the abstractive module to generate an abstractive summary. However, they use RL for the extractive module only. To the best of our knowledge, we are the first to utilize RL for

abstractive QfS.

We find an abundance of precedence of RL for abstractive summarization. Paulus et al. (2017) is the first work in employing RL to train LSTM models, without Teacher Forcing, for abstractive summarization. Pasunuru and Bansal (2018); Li et al. (2019) provide follow-up works with better reward functions. Both works train LSTM models without Teacher Forcing. Laban et al. (2020) were the first to employ RL to a Transformer architecture for abstractive summarization. They fine-tune the GPT2-small (Radford et al., 2019) model without Teacher Forcing. However, the omission of Teacher Forcing **led to a training time of 10 days for a generation length of just 10 tokens**. This highlights the severity of the problem which arises by omitting Teacher Forcing and using sequential generation during training, and motivates the need for our approach to effectively incorporate RL.

**Passage Embedding/Similarity:** A straight-forward way to obtain paragraph/passage embeddings is by application of some compositionality to the embeddings of the constituent words (Kenter et al., 2016; Hill et al., 2016; Sinoara et al., 2019; Iyyer et al., 2015). Yang et al. (2020); Jiang et al. (2019) attempt to generate passage embedding by training dual encoder networks to match similar documents. They rely on citation and recommendation networks to derive whether or not two passages are similar. Obtaining passage similarity/embedding is a common task in Dense Passage Retrieval (DPR) (Karpukhin et al., 2020). Ren et al. (2021) utilize Cluster Hypothesis (Jardine and van Rijsbergen, 1971) to enhance passage retrieval. However, they do not train any passage embedding network separately. Ginzburg et al. (2021) utilize pairwise sentence similarity to obtain similarity between two paragraphs. They use a noisy training dataset where paragraphs from the same document are considered similar and those from different documents are considered dissimilar. We are primarily motivated by the work of Vikraman et al. (2021). They use Cluster Hypothesis to group similar passages together in the DPR framework, for downstream tasks. Our work differs from Vikraman et al. (2021) in the following aspects: we explicitly train our model to output passage embeddings and we train on a much larger dataset.

# 3 Modeling

In this section we provide the formulation for QfS and Passage Embedding. Section 3.1 discusses the application of Policy Gradient to QfS. Section 3.2 discusses our approach to obtain passage embeddings. We present architectures in **Appendix G**.

## 3.1 Policy Gradient for QfS

We model the task using Reinforcement Learning (RL) framework. Reward Hypothesis (Sutton and Barto, 2018; Silver et al., 2021) states that all goals can be formulated as the maximization of a reward signal. Following this, our goal is to represent QfS using a reward adequately. Equations 1 and 2 specify the losses from Maximum Likelihood Estimation (MLE; used by Supervised Learning through cross-entropy loss) and Policy Gradient training. Comparing the two equations, it is visible that **MLE attempts to reward a token-by-token match** We take the RL path to generalize this rewarding mechanism using more sophisticated lexical and/or semantic rewards (Table 2). *Terminology*: $\mathbb{1}[x]$ evaluates to 1 if $x$ is *true* else 0, $y_k^*$ denotes the token generated at time-step $k$, $\tau$ is the trajectory obtained from the sequence of token generation, $\mathbf{q}$ and $\mathbf{d}$ denote the query and document representation respectively, and $n$ is the length of the generation.

$$\mathcal{L}_{\mathcal{MLE}} = -\sum_{t=1}^{n} \Big\{ \mathbb{1}[y_t = y_t^*] \log \mathcal{P}(y_t^*|y_1^*, y_2^*, \cdots, y_{t-1}^*, \mathbf{q}, \mathbf{d}) \Big\} \quad (1)$$

$$\mathcal{L}_{\mathcal{PG}} = -\sum_{t=1}^{n} \Big\{ (R(\tau) - b) \log \mathcal{P}(y_t^*|y_1^*, y_2^*, \cdots, y_{t-1}^*, \mathbf{q}, \mathbf{d}) \Big\} \quad (2)$$

We formulate QfS in the RL framework ($<\mathcal{S}$, $\mathcal{A}$, $\mathcal{R}$, $\mathcal{P}>$ tuple) as follows:

1. The agent (encoder-decoder network) exists in a state ($\mathcal{S}$), defined by the thus-far generated summary, the input query and the input document.

2. It takes action ($\mathcal{A}$) by sampling a token ($y_k^*$) from the probability over the vocabulary for the next time-step. The generation of probability distribution constitutes the state transition dynamics ($\mathcal{P}$).

3. The end of the episode (sequence of generation) is reached either through the generation of end-of-sequence token or by reaching the max episodic length.

4. At the end of the episode the agent achieves a reward ($\mathcal{R}$), which is then used by the Policy Gradient algorithm (Equation 2) to update the policy for the agent. We use the greedily decoded sequence to compute the baseline reward ($b$ in Equation 2), following Paulus et al. (2017).

However, it is intractable to run this vanilla RL framework for long episodes ($\sim$ 150 tokens, **Appendix G**, Table 10) when the policy has a huge number of parameters. It leads to a large time for forward and backward pass through the network, in addition to the large memory requirements. Refer to **Appendix E** for a space and time complexity analysis. We simulate the sequential action sampling through scheduled sampling for Transformers (Mihaylova and Martins, 2019). Following the strategy, we conduct a two-pass through the decoder: the second pass uses the sampled embeddings, obtained using Gumbel reparametrization trick (Goyal et al., 2017). This enables gradient backpropagation for the first pass through the decoder too. We find that using an implementation where gradients backpropagate through both passes leads to better results in automatic evaluation (Table 6). With this formulation, we train the policy network using a mixed objective loss (Equation 3, $\eta = 0.1$), following Paulus et al. (2017).

$$\mathcal{L}_{\mathcal{TOTAL}} = \eta \mathcal{L}_{\mathcal{MLE}} + (1 - \eta) \mathcal{L}_{\mathcal{PG}} \quad (3)$$

### 3.2 Passage Embedding and Semantic Similarity

Contemporary works rely on composing sentence/word embeddings to obtain passage embeddings. This forms an approximate representation of the generated and the ground truth summary. Thus, relying on these representations leads to an approximate notion of similarity. We amend that by creating a passage embedder using the **Cluster Hypothesis** (Jardine and van Rijsbergen, 1971), which states: *Passages that are clustered together answer to similar information needs*. According to the hypothesis, **summaries of the same query on the same document should be clustered together and have a high similarity score**. We use this insight to reward the generated summaries while

training, using the cosine similarity between the generated passage vectors. We find that similarity obtained from this embedding generation scheme leads to better results in human and automatic evaluation (Tables 6 and 8).

We use the Cluster Hypothesis for training our passage embedding model, using a dual encoder architecture (**Appendix G**). Our training scheme considers a special token (for example, *[CLS]* in BERT) to represent the passage. We calculate the similarity, $\hat{y}$, between passages, using the dot product of the embeddings ($\mathbb{E}_{\mathbf{p}}$ and $\mathbb{E}_{\mathbf{q}}$) produced by the dual encoder architecture, Equation 4. This causes similar $\mathbb{E}_{\mathbf{p}}$ and $\mathbb{E}_{\mathbf{q}}$ to lie closer in the embedding space. While training, we attempt to reduce the cross-entropy loss, Equation 5, between $\hat{y}$ and $y$ (the true labels; 1 for similar passages, else 0).

$$\hat{y} = \frac{1}{1 + e^{-\mathbb{E}_{\mathbf{p}} \cdot \mathbb{E}_{\mathbf{q}}}} \quad (4)$$

$$\mathcal{L}_{\mathcal{PE}} = -y \log \hat{y} - (1 - y) \log(1 - \hat{y}) \quad (5)$$

We use the trained passage embedding model to obtain embeddings for the generated summaries and the ground truth summaries in a batch while training the QfS model. We use cosine similarity as a reward signal in the RL framework.

## 4 Datasets

We use three datasets in this work: (*a*) ELI5 (Fan et al., 2019), (*b*) **R**eliable **QFS T**ester, RQFT (*our contribution*), and (*c*) **R**eddit **P**assage **E**mbedding **D**atase**T**, RPEDT (*our contribution*). In the following discussion, we provide a brief overview of the ELI5 dataset, and move on to discuss RQFT in Section 4.1, and RPEDT in Section 4.2.

Fan et al. (2019) proposed ELI5 for Longform Question Answering (LfQA). It consists of a triplets in the format: *<query, document, answer>*, where the query is a well-formed, open-ended question, and the answer is not a span from the document. Given the nature of the query and the answer, it fits perfectly in our problem statement. The dataset contains 234420, 9930 and 24820 samples for training, validation and testing respectively.

### 4.1 Reliable QFS Tester (RQFT) Dataset

We note two shortcomings of the ELI5 dataset:

1. Overlap of instances between training and validation set, reported by Krishna et al. (2021).

2. *<query, document, answer>* triplets in the ELI5 test set, where the document is irrelevant

| Reward | Description | Formula |
|---|---|---|
| ROUGE-L | Recall oriented reward to improve coverage | $\mathbb{ROUGE}_\mathbb{L}(GT, GN)$ |
| BLEU | Precision oriented reward to generate concise summaries | $mean_{i=1}^{4}(\mathbb{BLEU}_i(GT, GN))$ |
| SimCSE (Gao et al., 2021) | Semantic match obtained by averaging sentence embeddings | $cos(mean_m(\mathbb{E}_\mathbb{S}), mean_n(\mathbb{E}_\mathbb{T}))$ |
| SBERT (Reimers and Gurevych, 2019) | Semantic match obtained by averaging sentence embeddings | $cos(mean_m(\mathbb{E}_\mathbb{S}), mean_n(\mathbb{E}_\mathbb{T}))$ |
| SFPEG | Semantic match obtained using Passage Embedding | $cos(\mathbb{E}_{GT}, \mathbb{E}_{GN})$ |

**Table 2:** Various rewards used to train the RL models. We use both lexical and semantic rewards to promote lexical and semantic similarity in generation. SFPEG: Similarity From Passage EmbeddinG. $\mathbb{ROUGE}_\mathbb{L}$ and $\mathbb{BLEU}$ denote the standard ROUGE-L and BLEU functions; $mean$ denotes average; $cos$ denotes cosine similarity; $\mathbb{S}$ denotes a sentence from Ground Truth ($GT$) summary; $\mathbb{T}$ denotes a sentence from Generated ($GN$) summary.

to the query. We observed this shortcoming while analyzing the trained models on random samples from the test set.

Motivated by these, we curate a dataset with manual efforts, without any automation scripts, to ensure quality. We curate the dataset from two sources- (*i*) Wikipedia and (*ii*) high school textbooks. Both are excellent sources for a variety of concepts. We explain how the dataset is curated in **Appendix K**.

Table 3 presents statistics on the dataset. We can see that each document corresponds to more than 1 query on average. **This was an intentional decision made while curating the dataset to tackle topic centralization** (Baumel et al., 2016). We include an example of the dataset in the **Appendix** (Table 17). We establish the quality of the dataset in **Appendix L**.

While the size of RQFT is small, such a size for evaluation is not unprecedented for summarization (Angelidis and Lapata, 2018; Chu and Liu, 2019; Bražinskas et al., 2020; Amplayo et al., 2021; Angelidis et al., 2021). Inspite of the small size, our dataset acts as a high quality benchmark for QfS, covering 13 domains (listed in **Appendix K**), devoid of topic centralization.

### 4.2 Reddit Passage Embedding DataseT (RPEDT)

Cluster Hypothesis states: *Passages that are clustered together answer to similar information needs.* We gather RPEDT to facilitate the training of a passage embedding model using Cluster Hypothesis. Reddit forums (popularly known as subreddits)

| Characteristic | | Value |
|---|---|---|
| Size of dataset | | 250 |
| Avg. # of words | Query | 15.52 |
| | Document | 930.57 |
| | Summary | 115.72 |
| Avg. # of queries per document | | 1.41 |

**Table 3:** Statistics of RQFT dataset, we use NLTK to tokenize the strings.

contain data in the format of posts (queries) and comments (answers to the posts, often multiple in number). We scrape this data, forming a repository of queries and multiple answers to each query. We gather data from 39 subreddits, **Appendix F** describes the data gathering scheme and lists the subreddits. While training, we transform the dataset into the following format– <**p, q**>, where **p** and **q** are passages that may or may not answer the same query. We perform in-batch negative random sampling to generate **q** (one **q** per **p**) that does not answer the same query as **p**. Table 4 highlights the statistics; the last row denotes the total number of <**p, q**> samples generated during training. Our quality check experiment, **Appendix M**, validates the quality of RPEDT.

## 5 Experiments and Results

In this section we provide details on our experiments. Section 5.1 presents the results obtained on all the experiments. Finally, Section 5.2 presents the results obtained from human evaluation on RQFT. Table 5 lists all the trained models. **BART**

| Characteristic | Value | |
| --- | --- | --- |
| | Train | Test |
| # of questions | $154,722$ | $19,647$ |
| Avg. # of A per Q | 3.04 | 2.26 |
| Avg. # of W in Q | 15.58 | 17.80 |
| Avg. # of W in A | 145.36 | 170.88 |
| Training samples | $15,462,880$ | $223,046$ |

**Table 4:** Statistics of RPEDT, we use NLTK to tokenize into words. Q: Question, A: Answer, W: Words.

**SL** represents our implementation of of the model trained by Lewis et al. (2020) for ELI5.

| Model ID | Reward |
| --- | --- |
| **BART SL** | - |
| **BART R** | ROUGE-L |
| **BART R-SEM** | ROUGE-L + SimCSE |
| **BART R-B** | ROUGE-L + BLEU |
| **BART R-SBERT** | ROUGE-L + SBERT |
| **BART R-SFPEG** | ROUGE-L + SFPEG |
| **BART SFPEG** | SFPEG |

**Table 5:** Index of all the models trained in our work. All models except **BART SL** have been trained using Reinforcement Learning; the rewards are listed in the adjacent column. **BART SL** has been trained using Cross Entropy loss (Supervised Learning).

## 5.1 Automatic Evaluation Results

We perform three experiments- (*i*) **EXPT-I**: QfS on ELI5 test dataset, (*ii*) **EXPT-II**: QfS on RQFT, and (*iii*) **EXPT-III**: QfS on RQFT with random documents. Additionally, we also report automatic evaluation figures on DebatePedia (Nema et al., 2017). As the training and primary analysis involves only ELI5, we move the DebatePedia results to the Appendix (**Appendix D**). In EXPT-III, we replace the true document with another random document. The motivation behind this experiment was to rigorously test whether the trained models summarize the document based on the query, or generate tokens by ignoring the document altogether. We use the Fan et al. (2019) version of ELI5 dataset, not the KILT version. Hence we present competing results from Fan et al. (2019) and Lewis et al. (2020) only, and omit results on KILT version, such as those from Su et al. (2022).

We test the performance of the trained models using Beam Search (beam size = 15, minimum

tokens = 64, maximum tokens = 256). We find that using these generation conditions work best (based on automatic evaluation). We present the results generated for all the experiments in Table 6. We also present the average length of generations in Table 7. Using these generation parameters, we see that the BART SL model obtains better ROUGE-L (1.14 points) than the one presented by Lewis et al. (2020). We attribute this gain to the usage of Scheduled Sampling, leading to more robustness in generation than a model trained using Teacher Forcing.

We can see in Table 6 that the models trained using Reinforcement Learning (RL) perform significantly better than BART SL (the baseline), with a **10.62 improvement** on ROUGE-L for ELI5 test set. We can also see that the models obtain significantly better results, than the baseline, on our test dataset (EXPT-II), achieving as high as **14.89 improvement** on ROUGE-L. The RL-based models achieve better scores on our dataset as compared to ELI5 test set, however, we fail to see such significant gains for BART SL. We present a possible reason for this in Section 6.3. Also, we see that the scores for EXPT-II and EXPT-III are closer for BART SL, in comparison to the RL-based models. This indicates two things-

1. BART-SL model generates relevant (to the query) content (irrespective of the document-random or true). But, the RL models actually utilize the provided document. This shows that the RL models truly learn the task of QfS, justifying the significant boost ($\sim 10$ points).

2. While Krishna et al. (2021) point out several shortcomings of the ELI5 dataset, it can be seen that, using the dataset, RL models can learn to summarize according to a query given the true document.

## 5.2 Human Evaluation Results

We present human evaluations on Fluency and Correctness of our models. We use two annotators for human evaluation. We ask them to rate the Fluency on a Likert Scale ("*Very Poor*", 0 to "*Very Good*", 4). For Correctness, we follow a YES (1) or NO (0) marking scheme. We randomly sample 50 instances from the RQFT dataset, and present the generated summaries, and the human written summaries to the annotators. The summaries are anonymized, such that the evaluators have no understanding of the source: human written or auto-

| Model | ELI5 dataset | | | Our dataset (EXPT-II) | | | Our dataset (EXPT-III) | | |
|---|---|---|---|---|---|---|---|---|---|
| | R-1 | R-2 | R-L | R-1 | R-2 | R-L | R-1 | R-2 | R-L |
| Fan et al. (2019) | 28.9 | 5.4 | 23.10 | 28.82 | 6.97 | 25.41 | 23.13 | 4.39 | 20.91 |
| Lewis et al. (2020) | 30.6 | 6.2 | 24.3 | - | - | - | - | - | - |
| BART SL (b) | 29.68 | 5.89 | 25.44 | 29.67 | 7.88 | 26.40 | 23.32 | 4.51 | 20.96 |
| BART R | 38.93 | 8.05 | 34.54 | 43.08 | 15.30 | 39.08 | 24.38 | 4.40 | 22.17 |
| BART R-SEM | 38.02 | 6.56 | 33.13 | 44.12 | 15.15 | 40.39 | 25.39 | 4.51 | 23.06 |
| BART R-B | **39.52** | **8.25** | **34.92** | 43.46 | 15.67 | 39.29 | 26.01 | 4.88 | 23.45 |
| BART R-SBERT | 36.9 | 6.36 | 32.8 | 42.93 | 15.10 | 39.56 | 25.41 | 4.47 | 23.19 |
| BART R-SFPEG | 39.40 | 6.92 | 34.10 | **45.52** | **16.83** | **41.29** | 25.89 | 4.99 | 23.50 |
| BART SFPEG | 34.76 | 5.96 | 29.66 | 44.30 | 15.79 | 40.39 | 25.82 | 4.63 | 23.61 |

**Table 6:** Quantitative comparison of our models. For the purposes of comparison, we also provide the results obtained by Fan et al. (2019) and Lewis et al. (2020). We use ROUGE metrics, ROUGE-1 (R-1), ROUGE-2 (R-2), and ROUGE-L (R-L) to compare the models on both the ELI5 test dataset (EXPT-I) and RQFT (EXPT-II: with true document and EXPT-III: with random document). We use BART SL as the baseline (denoted by b) to judge the efficacy of training models under the Reinforcement Learning framework. BART SL is our implementation of Lewis et al. (2020), hence row 3 represents the results for Lewis et al. (2020) for EXPT-II and EXPT-III.

| Model | EXPT-I | EXPT-II |
|---|---|---|
| BART SL | 77.32 | 95.78 |
| BART R | 234.29 | 234.02 |
| BART R-SEM | 239.98 | 233.84 |
| BART R-B | 241.27 | 235.89 |
| BART R-SBERT | 237.09 | 235.19 |
| BART R-SFPEG | 239.23 | 232.73 |
| BART SFPEG | 239.19 | 234.93 |

**Table 7:** Length of generations (# of words) from the models. We use NLTK to tokenize the generations. We provide an analysis on how RL models utilize more tokens for better summarize in Section 6.1.

matically generated or even the generator model. Table 8 presents the results generated from human evaluation. We can see a similar trend as in Table 6: BART R-SFPEG model obtains the highest Fluency and Correctness.

We also compute the IAA scores for Fluency and Correctness for the evaluations of all the models. For Fluency, we map the scores to ratings: *Positive* (3 and 4), *Neutral* (2) and *Negative* (0 and 1), and compute the fraction of times the annotators agree on the ratings. We observe that the annotators agree almost always, with a minimum agreement of 0.88 (for BART SL). For Correctness, we use Cohen Kappa score to report the agreement. We observe *moderate* agreement (McHugh, 2012), with a minimum score of 0.61 (for BART R-SEM). Through hypothesis testing, we validate that BART

R-SFPEG model is significantly better than the BART SL model. We find that BART R-SFPEG is significantly more correct than BART MLE, with a $10\%$ significance level ($p\ value\ =\ 0.077$). In terms of Fluency, BART R-SFPEG is significantly more fluent than BART SL model with a $1\%$ significance level ($p\ value\ =\ 0.00012$).

| Model | Fluency | Correctness |
|---|---|---|
| BART SL | 3.08 / 3.12 | 0.26 / 0.28 |
| BART R | 3.34 / 3.30 | 0.38 / 0.32 |
| BART R-SEM | 3.26 / 3.23 | 0.24 / 0.18 |
| BART R-B | 3.24 / 3.26 | 0.28 / 0.26 |
| BART R-SEM | 3.28 / 3.27 | 0.22 / 0.26 |
| BART R-SFPEG | **3.46 / 3.48** | **0.48 / 0.44** |
| BART SFPEG | 3.06 / 3.04 | 0.28 / 0.26 |
| Human | 4.0 / 3.98 | 1.0 / 1.0 |

**Table 8:** Results from human evaluations of the machine generated text. The scores are reported in A / B format; where A is the average score computed from ratings by Annotator 1 and B is the average score computed from ratings by Annotator 2.

## 6 Analysis

Our initial analyses on the ELI5 test set revealed a recurring problem: our models (all of the 7 models) would copy the query into the generated summary. On further probing, we discovered that this correlated highly with the absence of content in the

document relevant to the given query. Frequent occurrences motivated us to create a manually curated test set for reliable analyses. In this section we present a detailed analysis of our models. In Section 6.1 we present a comparative analysis of all the models, in Section 6.2 we analyse the models when multiple queries are posed on the same document and in Section 6.3 we analyse our models when the document is replaced with a random document. Our analysis reveals that the RL models actually learn QfS as compared to the BART-SL model, which has the tendency to ignore the provided document. This justifies the significant boost ($\sim$ 10 points) in performance.

## 6.1 Comparative Analysis

All the Reinforcement Learning (RL) based models generate self-contained, easy to follow summaries. On the other hand, the BART SL model generates very crisp summaries, too frugal with the amount of words (Table 7). The RL-based models present a lot of relevant material in the generated summary, but the BART SL model jumps straight to the point. While the latter can be usually desired, the tendency to be too frugal with words can leave the user wanting for more. We include an example in **Appendix A** to highlight the differences. We believe that this difference is a result of using ROUGE as a reward signal. ROUGE leads the model to choose more of the relevant content while training to increase the reward. However, we also observe a downside to this: as ROUGE rewards same word stems, we see multiple tokens with the same stem, discussed in **Appendix A**. Although the BART-SFPEG model does not use ROUGE as a reward, we see that it also has verbose generations. This is also understandable: choosing more relevant content increases semantic match too.

We also note that BART SL model hallucinates (Ji et al., 2022) significantly more than all the RL-based models, inspite of the crispness, which contributes to lower Correctness score (Table 8). We highlight this phenomenon in **Appendix A**.

## 6.2 Multiple Queries Same Document

We also test abilities of the models when they are probed with different queries on the same document. We observe that the Reinforcement Learning (RL) based models perform better in this arena too. **Appendix B** includes an example highlighting the difference between the BART SL and an RL-based model. We see that, although the BART SL model

manages to generate different summaries for the two queries, they are either hallucinated or mostly unrelated to the query. Whereas, the RL-based model manages to understand the necessities of the queries, and generates summaries for them.

## 6.3 Analysis for Random Document

Section 5.1 motivates the need for this experiment. Quantitative results show that BART SL does not have as significant a drop in performance as the Reinforcement Learning (RL) based models when we use a random document. While analyzing the generations, we observe that the BART SL model ignores the document. Although the generated text stays relevant to the query, it cannot be stated as a query focused summary of the document, as the content is absent in the document. This leads us to believe that the BART SL does not truly learn to solve QfS, which also explains the relatively insignificant change in scores when the quality of the dataset is improved (ELI5 test set vs RQFT, Table 6).

We include an example of generation from random document for BART SL and an RL-based model in the **Appendix C**. We see that the RL-based model generates content from the document, based on whatever it understands from the query, which strongly confirms our belief that the RL-based models learn a much better solution to QfS.

## 7 Conclusion and Future Work

In this work we present Reinforcement Learning (RL) based models for the task of QfS. We observe that these models perform significantly better than a model, with the same architecture, trained using Supervised Learning (SL). We also observe that the RL-based models generate summaries with much lesser hallucination than the Supervised Learning model. Additionally, we also resolve the conflict of employing RL in Transformers with Teacher Forcing, by utilizing Scheduled Sampling. We present a novel reward derived using the Cluster Hypothesis. Through the work, we also contribute two datasets for the community: RPEDT, to train passage embedding models, and RQFT, a gold standard test dataset for analyses of QfS models.

The takeaway from our work is that RL is a better framework to tackle QfS. We observe that RL helps the model learn QfS much better than SL, even with straightforward lexical rewards, such as ROUGE. Additionally, we also conclude, from results, that

Cluster Hypothesis leads to much better semantic feedback than competing passage embedders.

Training RL models for long horizon (higher generation length) poses challenges in terms of exploration of action space and temporal credit/blame assignment (Jiang and Agarwal, 2018). In our future work, we would focus on tackling this challenge. Our motivation is that it would reduce the convergence time, and provide even better results.

## 8 Ethics Statement

We present two new datasets in our work: RPEDT and RQFT. While curating RQFT, we attempt to refrain from including any sensitive topic into our dataset, based on the knowledge of the curators and the quality check annotators. However, we cannot guarantee such claims for RPEDT. We understand that the passage embedding models trained on the dataset can learn biases and stereotypes present in RPEDT. Hence, we urge the community to use our passage embedding models and RPEDT with caution.

We utilize a total of 4 annotators to conduct quality check experiments and human evaluation. 2 annotators are post-graduates and the other 2 are under-graduates in the Computer Science department of an organization (medium of communication: English). The annotators have been paid sufficiently, according to the geographical location, on an agreed upon rate.

## 9 Limitations

The most striking limitation of our work is the necessity of huge computation power, which has restricted us from an extensive experimentation on hyperparameter search. In light of such limitations, we report results from a single run. And we also acknowledge the presence of hyperparameter configurations that can lead to better performance of the models. However, we note that such a limitation does not mirror during inference, and our models can be cheaply (relatively) deployed for public use.

Another limitation of our work is that the passage embedding models have been trained on relatively limited data gathered from only one source (Reddit). We acknowledge the fact that training on varied sources can lead to better representation of passages, however, keeping time and resource constraints in mind, we train our models on data gathered from only one source.

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

## A  Comparison of Generations from the Models

We present a randomly sampled instance from our test dataset in Table 18: we include the query and the generations from all six models, we omit the document for readability.

We can see that the BART SL model generates a very crisp response to the query, stating what population growth is. However, it fails to respond to the other aspect of the query: *What does population change indicate for an area?* It also moves on to generate an ill-stated hallucinated (Ji et al.,

2022) example (*absent in the document*). We note the absence of such hallucination in the Reinforcement Learning (RL) models. However, we note that only 3 of the 5 RL-based models are able to incorporate both aspects of the query. We also note digressions in the RL-based models, with BART R-SEM generating significantly more unrelated content. We observe that this is a general trend: BART R-SEM often generates content very haphazardly. We hypothesize that this is caused by using an approximate semantic similarity scheme as a reward signal.

We also note repeated generations (not necessarily consecutive) with same word stems (such as *percent*, leading to repeated *percentage* in the BART R-SFPEG generation) in the RL-based models, trained using ROUGE as a reward.

## B  Comparison of Models for Multiple Queries on Same Document

We present two randomly sampled instances, two queries with the same document, from our dataset in Table 19: we include the query and the generations from BART SL and BART R model only, we omit the document for readability.

For the first query, BART SL hallucinates that *Green Revolution* has not helped India, given that the document states otherwise. BART R model picks up the key arguments from the document and indeed presents a decent summary pertaining to Green Revolution's help to India.

For the second query, BART SL ignores the query altogether and generates some related sentences only. While BART R picks up key information related to *Buffer Stock* and articulates a satisfactory summary again.

## C  Comparison of Models for QfS with Random document

We present two random samples from our test set where the documents have been replaced by another random, unrelated document, in Table 20. We see that for the given query, BART SL and BART R have unrelated (and different) documents. BART SL manages to generate content very much related to the query, which indicates an unfair strong influence of the query on the generation, so much so that the document gets ignored. However, BART R generates content from the document indicating that document indeed has appreciable influence on its generation, which is the ideal case in QfS.

## D Results on DebatePedia

We generate result on the test set of DebatePedia using our best RL model: BART R-SFPEG. We utilize the model trained on ELI5 directly to generate results on the test set of DebatePedia. We observe that we perform atleast as good as models trained/fine-tuned on DebatePedia, proving the generalization of our approach.

| Model | R-1 | R-2 | R-L |
|---|---|---|---|
| BERTABS(Liu and Lapata, 2019) | 13.3 | 2.8 | 2.8 |
| BART (Lewis et al., 2020) | 21.4 | 6.3 | 18.4 |
| LQSUM (Xu and Lapata, 2022) | 23.5 | **7.2** | 20.6 |
| BART R-SFPEG | **24.2** | 6.9 | **20.7** |

**Table 9:** Results on the DebatePedia test set. R-1: ROUGE-1, R-2: ROUGE-2, R-L: ROUGE-L

## E Time and Space Complexity Analysis

We present an analysis for the attention layer only, as that is the bottleneck for space utilization and time consumption, as input length increases. An $L$-layered decoder utilizes $\mathcal{O}(n^2)$ time and space for predicting the next token provided the previous $n$ tokens. If the decoding is done without Teacher Forcing, that is $n^{th}$ token is generated and then passed through the decoder as an input to generate the $(n + 1)^{th}$ token, then the space and time required is:

$$\mathcal{O}(1^2 + 2^2 + 3^2 + \cdots + n^2) = \mathcal{O}(n^3)$$

On the other hand, if we employ the two pass decoding using Scheduled Sampling, the time and space utilized is $\mathcal{O}(2 * n^2) = \mathcal{O}(n^2)$.

## F RPEDT Scraping Scheme

We generate RPEDT by scraping posts (queries) and comments (answers) from 39 subreddits. Table 21 presents the list of the subreddits we use to generate the dataset. We gather data within the timeframe: *July, 2011* and *July, 2022*. We set the following criteria to filter the gathered posts and comments:

1. We consider a post (and related comments)

gatherable if and only if the post has a score[3] of atleast 2.

2. We consider a comment (under a gatherable post) if and only if the comment has a score of atleast 2.

After scraping the dataset, we discard posts with no associated comments and comments with less than 50 words[4]. Finally, we divide the dataset into train and test sets as follows: we accumulate all the posts (and associated comments) from the *explainlikeimfive* subreddit into the test set, and use the rest for training.

## G Architecture Details

**QfS:** We train a Sequence-to-Sequence model to generate summaries, given the query and the document. Our trained model uses BART (Lewis et al., 2020) as the backbone, to obtain representations of the query and document, and finally generate the summary. Figure 1 illustrates the input to and output from the architecture. Query and document are concatenated and fed to the encoder, and we expect the decoder to generate the summary. We use pretrained BART-large model as the starting point for our QfS model.

Table 10 reports statistics on the length of inputs and outputs. Accordingly, **we increase the positional embeddings' length** of the BART model in our implementation to **1568** (approximately equal to $\mu_{valid} + 2\sigma_{valid}$, where $\mu$ and $\sigma$ denote average and standard deviation of the encoder input length respectively, Table 10). The first 1024 places of the new positional embeddings are initialized using the positional embeddings of the pre-trained BART model.

**Passage Embedding:** We train an encoder-only model to generate passage embeddings. Specifically, we fine-tune the BERT-large model (Devlin et al., 2019) to output passage embeddings through the *[CLS]* token. Figure 2 depicts the dual encoder scheme used to fine-tune the BERT model. We use Equation 4 and 5 to train the model. In addition to the passage similarity task, we also employ Masked Language Modelling (MLM) task during fine-tuning. We observed that the fine-tuned model experienced an increased validation perplexity for MLM, as compared to the pretrained BERT-large

---

[3]We compute score as the difference between the upvotes and downvotes, $score = upvotes - downvotes$

[4]We use SpaCy to tokenize the comments.

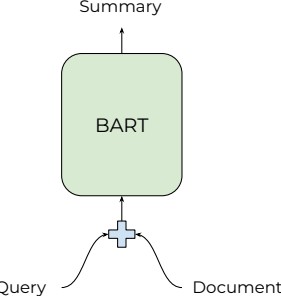

**Figure 1:** Architecture for the Query focused Summarizer.

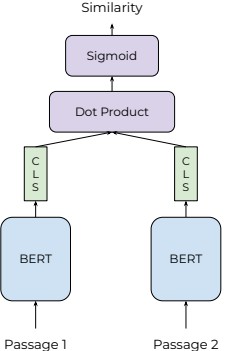

**Figure 2:** Dual Encoder schematic for training the Passage Embedding model.

model. We mirror the schemes for MLM training suggested by Devlin et al. (2019), during fine-tuning. We keep the default maximum length of BERT (512 tokens), as we see that the average generations from the QfS model are expected to be $\sim 150$ tokens.

## H  Training Details for QfS

We use the BART-large (12 layers, 16 attention heads, 1024 dimensional embedding) pretrained model as our starting point. We train on 8 x A100 machines, with all the models taking a total time of 21 days to be trained till convergence. Table 11 outlines the hyperparameters used for training. We train all the models without any explicit learning rate scheduler, and found this to lead to faster convergence in our experiments.

## I  Training Details for Passage Embedding

We fine-tune both BERT-base and BERT-large (both cased) pretrained models. We train on 2 x A100 machines, with the base model taking a total of 64 hours and the large model taking a total of 186 hours. Table 12 lists the hyperparameters used

| Split | Statistic | Average | Std. Dev. |
|-------|-----------|---------|-----------|
| Train | Enc Inp Len | 1120.06 | 260.14 |
|       | Dec Out Len | 149.07 | 167.09 |
| Valid | Enc Inp Len | 1122.85 | 250.48 |
|       | Dec Out Len | 147.08 | 165.55 |
| Test  | Enc Inp Len | 1123.07 | 255.31 |
|       | Dec Out Len | 147.24 | 162.62 |

**Table 10:** Statistics of the lengths of inputs (Enc Inp Len) to the encoder (query + document, concatenated) and outputs (Dec Out Len) from the decoder (summary). The statistics are obtained after tokenization using BART tokenizer.

| Hyperparameter | Value |
|----------------|-------|
| Max Epochs | 5 |
| Learning Rate | $2^{-5}$ (SL) / $5^{-7}$ (RL) |
| Effective Batch Size | 128 |
| Max Sequence Length | 1568 |
| Optimizer | Adam |

**Table 11:** Values of various hyperparameters used while training QfS models. Effective batch size is the number of samples passed through the model before parameter updates. SL: Supervised Learning, RL: Reinforcement Learning.

for training. We use the combined loss (Passage Similarity loss, Equation 5, + MLM loss) to train the models. We observe that models fine-tuned for a single epoch perform best on the test set of RPEDT. Appendix J presents a comparison of the two trained models, in terms of the downstream task performance.

| Hyperparameter | Value |
|----------------|-------|
| Max Epochs | 2 |
| Learning Rate | $1^{-6}$ |
| Effective Batch Size | 4096 |
| Max Sequence Length | 512 |
| Optimizer | Adam |

**Table 12:** Values of various hyperparameters used while training Passage Embedding model. Effective batch size is the number of samples (positive labelled instances + in-batch sampled negative labelled instances) passed through the model before parameter updates.

## J  Bert-large vs BERT-base for Passage Embedding

We compare BERT-large and BERT-base, as passage embedding models, based on the quality of the models in the downstream task of QfS. We use the fine-tuned passage embedding models to reward generated summaries, on how similar they are to the ground truth summaries, while training the QfS model using Reinforcement Learning. Table 13 presents the results obtained from two QfS models: BART R-SFPEG-L (reward: ROUGE + SFPEG from the BERT-large model) and BART R-SFPEG-B (reward: ROUGE + SFPEG from the BERT-base model). We observe that BERT-large led to significantly better results, and hence proceeded to use BERT-large fine-tuned passage embedding model for further experiments, such as human evaluation, in our work.

| Experiment | Model | Metrics | | |
|---|---|---|---|---|
| | | R-1 | R-2 | R-L |
| EXPT-I | M-I | 39.52 | 6.92 | 34.92 |
| | M-II | 38.74 | 6.27 | 33.50 |
| EXPT-II | M-I | 45.52 | 16.83 | 41.29 |
| | M-II | 43.94 | 15.42 | 40.06 |

**Table 13:** Comparison of performance of BART R-SFPEG-L (M-I) and BART R-SFPEG-B (M-II) for EXPT-I and EXPT-II.

## K  Details on RQFT Curation and Domains

We curate the dataset from two sources- (*i*) Wikipedia and (*ii*) high school textbooks. While preparing samples from Wikipedia, we manually copy text from the website, form queries on the text data, and write a summary based on the query. High school textbooks contain chapters followed by open-ended questions on concepts within the chapters. We use the questions at the end of the chapter to form our dataset. The answers to the questions are also freely available from online sources, which aid students in their exams. We create <*query*, *document*, *summary*> triplets by (*a*) Copying material from the chapter (relevant to the query) and (*b*) Copying the answer to the question from an online source (the curator verifies whether or not the answer is correct by reading the question and the extracted document). While following this process, we noted that multiple questions could be answered from consecutive paragraphs of the chapters. Using this, we copied material relevant to multiple queries from the chapter to form a single document, ensuring absence of topic centralization in RQFT (Baumel et al., 2016). As these were consecutive paragraphs, coherence was trivially ensured.

We construct the dataset from Wikipedia and High School textbooks. The domains from Wikipedia involve: POLITICS, FAMOUS PERSONALITY, COUNTRY, VIDEO GAME and TV SERIES. The domains from High School textbooks include SCIENCE, SOCIAL SCIENCE, BIOLOGY, BUSINESS STUDIES, GEOGRAPHY, HISTORY, POLITICAL SCIENCE and SOCIOLOGY. Effectively the dataset includes samples from 13 domains.

## L  Quality Check: RQFT

We report quality check results performed by two independent annotators. We design the quality check framework as follows:

1. Each annotator is presented with 30 <*query*, *document*, *summary*> triplets, of which 10 samples (*random-samples*) are such that the summary does not correspond to the query and document. We keep 5 from *random-samples* and 10 from the rest 20 (*correct-samples*) common between the two annotators to compute the Cohen Kappa score.

2. Each annotator is asked to mark **YES (score 1)** or **NO (score 0)** to the following questions:

   (a) "*Is the query relevant to the document?*" (ANN-1): This ensures that the dataset contains queries that are synchronous with the paired document. The ideal score is 1 for both *random-samples* and *correct-samples*.

   (b) "*Does the summary answer the query while being faithful to the document?*" (ANN-2): This ensures that the written summary is indeed correct and is not hallucinated (Ji et al., 2022). The ideal score is 0 for *random-samples* and 1 for *correct-samples*.

   (c) "*Is the summary grammatically correct, fluent, and coherent?*" (ANN-3): This ensures the readability, fluency, and

grammatical correctness of the written summary. The ideal score is 1 for both *random-samples* and *correct-samples*.

Table 14 presents the results obtained from the quality check experiment. The values represent the average score the annotators assigned for the annotations: ANN-1, ANN-2, and ANN-3, over the 30 samples. Using the 15 common samples, we observe a perfect Cohen Kappa (1.0) between the annotators.

| | ANN-1 | ANN-2 | | ANN-3 |
| --- | --- | --- | --- | --- |
| | | + | − | |
| Annotator 1 | 1.0 | 0.95 | 0.0 | 0.95 |
| Annotator 2 | 1.0 | 1.0 | 0.0 | 1.0 |

**Table 14:** Quality scores assigned by annotators 1 and 2 for ANN-1, ANN-2, and ANN-3. +: *correct-samples*, −: *random-samples*.

## M  Quality Check: RPEDT

In order to validate the quality of RPEDT, we provide <$\mathbf{Q}$, $\mathbf{P_1}$, $\mathbf{P_2}$> triplets to annotators, where $\mathbf{P_1}$ and $\mathbf{P_2}$ answer the query, $\mathbf{Q}$. We ask them to rate how well does each passage ($\mathbf{P_1}$ and $\mathbf{P_2}$) answer the query, on a scale of 0 (*bad*) to 2 (*good*). We employ two annotators for the task, who rate a total of 707 triplets (79 common). Table 16 presents the results from annotation. We measure agreement by noting the number of times they rate passages with the same score. We observe that annotators agree 78.4% on the rating for $\mathbf{P_1}$ and 86% on the rating for $\mathbf{P_2}$.

| | $R_{\mathbf{P_1}}$ | $R_{\mathbf{P_2}}$ |
| --- | --- | --- |
| Annotator 1 | 1.79 | 1.73 |
| Annotator 2 | 1.87 | 1.86 |

**Table 16:** Average ratings provided by the two annotators for $\mathbf{P_1}$ ($R_{\mathbf{P_1}}$) and $\mathbf{P_2}$ ($R_{\mathbf{P_2}}$), higher is better.

| Score | Criteria |
| --- | --- |
| 0 | Passage is not related to the topic, or does not discuss anything relevant to the question |
| 1 | Passage partially answers the question |
| 2 | Passage completely answers the question |

**Table 15:** Annotation guidelines for the quality check of RPEDT.

**Query**: Why do stars twinkle? And, why don't planets twinkle?

**document**: The twinkling of a star is due to atmospheric refraction of starlight. The starlight, on entering the earth's atmosphere, undergoes refraction continuously before it reaches the earth. The atmospheric refraction occurs in a medium of gradually changing refractive index. Since the atmosphere bends starlight towards the normal, the apparent position of the star is slightly different from its actual position. The star appears slightly higher (above) than its actual position when viewed near the horizon. Further, this apparent position of the star is not stationary, but keeps on changing slightly, since the physical conditions of the earth's atmosphere are not stationary, as was the case in the previous paragraph. Since the stars are very distant, they approximate point-sized sources of light · · ·

**Summary**: Stars appear to twinkle due to atmospheric refraction. The light of a star after the entry of light into the earth's atmosphere undergoes refraction continuously till it reaches the surface of the earth. Stars are far away. So, they are a point source of light. As the path of light coming from stars keeps changing, thus the apparent position of stars keeps changing and the amount of light from stars entering the eye keeps twinkling. Due to this, a star sometimes appears bright and sometimes dim, which is the effect of twinkling. The planets are much nearer to the earth than the stars and because of this, they can be considered as a large source of light. If a planet is considered to be a collection of a very large number of point sources of light, then the average value of change in the amount of light entering the eye from all point-size light sources is zero. Due to this the effect of twinkling is nullified.

**Table 17:** Example of a randomly sampled *<query, document, summary>* triplet from the annotated dataset.

| **Query**: What is population growth? What does population change indicate for an area? |
|---|
| **BART SL**: Population growth refers to the change in number of inhabitants of a territory during a specific period of time . It can be expressed either in absolute numbers or in terms of percentage percentage of the total population . ==For example , let 's say you have a region with a population of 1, 000 people . Over the next 10 years , there will be an increase in population growth as more people move in and out of that area . This means that the population will increase by 10 %== . |
| **BART R**: Population change refers to the change in number of inhabitants of a territory during a specific period of time. This change may be positive as well as negative. Population growth occurs not only by increasing births rate but also due to decreasing death rate. There are two ways by which the population size changes – births and deaths. The crude birth rate (CBR) is a simple method of measuring the rate of births in a particular region. It can be expressed either in terms of absolute numbers or in numbers of live births per thousand of population. Similarly, the mortality rate is an important indicator of the region's demographic structure, social advancement and levels of its economic development. |
| **BART R-SFPEG**: Population change refers to the change in number of inhabitants of a territory during a specific period of time. This change may be positive as well as negative. Population growth occurs not only by increasing births rate but also due to decreasing death rate.Population change is an important indicator of economic development, social upliftment and historical and cultural background of the region. It can be expressed either in absolute numbers or in terms of percentage. ==Percentage== is the ==percentage== of population in a particular geographical area that is increasing or decreasing over a time period as a function of its demographic structure, socio-economic advancement and other factors such as its geographical location, population density, age, sex ratio, etc. (%)Population growth is a dynamic process that affects the population size over time as it changes from one place to another. |
| **BART SFPEG**: Population change refers to the change in number of inhabitants of a territory during a specific period of time. This change may be positive as well as negative. Population growth occurs not only by increasing births rate but also due to decreasing death rate.Population change is an important indicator of economic development, social upliftment and historical and cultural background of the region. It can be expressed either in absolute numbers or in terms of percentage. There are two ways by which the population size changes – births, deaths and migration. The crude birth rate (CBR) is expressed as the ratio of live births in a year per thousand of population. Conversely, the mortality rate is a simple method of measuring mortality of any area. |
| **BART R-SEM**: By and large mortality rates are affected by the region's demographic structure, social advancement and levels of its economic development. Population growth occurs not only by increasing births rate but also due to decreasing death rate. · · · When people move from one place to another places, their population may increase or decrease depending on their place of origin and the place they move to is called the Place of destination. Population change refers to the change in the total population of a territory during a specific period of time. Population change is an important indicator of economic and social development, socio-economic status and historical and cultural background of the territory. |
| **BART R-B**: Population change refers to the change in number of inhabitants of a territory during a specific period of time. This change may be positive as well as negative. Population growth occurs not only by increasing births rate but also due to decreasing death rate. There are different ways by which the population size changes in a region depending on the region's demographic structure, social advancement and levels of its economic development. It can be expressed either in absolute numbers or in terms of percentage. The crude birth rate (CBR) is a simple method of measuring mortality of any area and it is expressed in numbers of live births per thousand of population for a particular region. |

**Table 18:** Comparison of generations from all the trained models. The query has two aspects, color-coded by teal and violet. Text-segments in generations are colored accordingly based on which part of the query they answer.

| **Query One**: How has green revolution helped India, if at all? | **Query Two**: Why is a buffer stock created by the government? |
| --- | --- |
| **BART SL**: I do n't think the green revolution has helped at all . The food insecure people are disproportionately large in some regions of the country because of poverty , tribal and remote areas , regions more prone to natural disasters etc . In fact , the states of Uttar Pradesh (eastern and south - eastern parts of India ) , Orissa , Andaman and Nicobar Islands , and West Bengal have been experiencing famine since the 1970 's . | **BART SL**: If there is a shortage of food in a region , the government can buy it from the market at a lower price than what people are willing to pay for it , and then distribute it to the people who need it . This way , they do n't have to worry about people going hungry all the time . |
| **BART R**: The food insecure people are disproportionately large in some regions of the country, such as economically backward states with high incidence of poverty, tribal and remote areas, regions more prone to natural disasters etc. After Independence, Indian policymakers adopted all measures to achieve self-sufficiency in food grains. India adopted a new strategy in agriculture, which resulted in 'Green Revolution', especially in the production of wheat and rice. The farmers are paid a pre-announced price for their crops. This price is called Minimum Support Price (MSP). The MSP is declared by the government every year before the sowing season to provide incentives to farmers for raising the productivity of these crops and to encourage them to increase their production in order to reduce their dependence on imported foodgrains (such as wheat, rice, maize, sugar, pulses, oilseeds etc.). In the mid-1960s, wheat was the most productive crop in India and it was grown in a large number of small and medium-scale farms in rural areas and in small towns and villages. These small farms were able to produce large quantities of foodgrain and the farmers were rewarded with a subsidy of Rs 1,000 per acre per hectare | **BART R**: Since the advent of the Green Revolution in the early-1970s, the country has avoided famine even during adverse weather conditions. India is aiming at Self-sufficiency in Foodgrains since Independence. The attainment of food security therefore involves eliminating current hunger and reducing the risks of future hunger. After Independence, Indian policymakers adopted all measures to achieve self-sufficient in food grains. This system has two components: (a) buffer stock, and (b) public distribution system.Buffer Stock is the stocks of grains procured by the government through the Food Corporation of India (FCI). The FCI purchases wheat and rice from the farmers in states where there is surplus production. These grains are sold to the consumers at a price lower than the market price of these grains, which is known as the Minimum Support Price (MSP). In the case of wheat or rice, it is sold at the MSP of Rs 2,000 per tonne and in case for rice at Rs 1,500/tonne of rice the price is Rs 3,200/towards Rs 4,300/tons for the same amount of grain. There are a buffer price to be paid to farmers for procuring the grains at these prices. |

**Table 19:** Comparison of BART SL and BART R in the case of multiple queries over the same document. Text in red highlights the hallucinated segment in the generation.

| Query: Why do organisms need water? |
|---|

| Document: Primary activities include agriculture, animal husbandry, forestry, fishing, mining and quarrying, etc. Secondary activities include manufacturing industry, building and construction work, etc. Tertiary activities include transport, communications, commerce, administration and other services. The proportion of people working in different activities varies in developed and developing countries. Developed nations have a high proportion of people in secondary, and tertiary activities. Developing countries tend to have a higher proportion of their workforce engaged in primary activities. ⋯ |
|---|

| BART SL: Water is an essential part of the life cycle of every living thing . It 's what keeps them alive and allows them to grow and reproduce . Without it , they would n't be able to do all the things they do . For example , water is essential for photosynthesis , which is the process of turning food into energy that can be used by other organisms . |
|---|

| Document: During 1998, tropical cyclones formed within seven different tropical cyclone basins, located within various parts of the Atlantic, Pacific, and Indian Oceans. During the year, a total of 125 tropical cyclones had formed, with 72 of them have been named by various weather agencies when they attained maximum sustained winds of 35 knots (65 km/h, 40 mph). The strongest tropical cyclones of the year are Zeb, Ron and Susan which peaked with a pressure of 900 hPa (26.58 inHg). Hurricane Mitch of late ⋯ |
|---|

| BART R: The most notable storms were Hurricane Georges and Hurricane Mitch. Four Category 5 tropical cyclones were formed in 1998. Although a semi-permanent anticyclone persisted through the summer of 1998, causing most of the storms to remain at sea, some storm did threaten the Baja California Peninsula due to a weak ridge of high pressure in the mid-latitudes, and the subtropical low pressure area over the central and eastern Pacific Ocean (Pacific Oscillation (POW)). However, the most powerful storms that formed during the season were the hurricanes and tropical depressions (tropical storms). The most destructive storms of this year were hurricanes, which caused significant damage and killed at least 11,000 people, making the system the second deadliest Atlantic hurricane in recorded history, behind only the Great Hurricane of 1780. In the North Atlantic basin, it was the deadliest hurricane season on record, with the damages amounting to $9.37 billion and $6.08 billion (1998 USD) in damage, respectively. The season had above average activity, featuring a record-breaking 30 tropical storms, as well as a high number of hurricanes with a maximum sustained winds of more than 111 mph (178 km/h). |
|---|

Table 20: Comparison of BART R and BART SL for QfS from random document.

| explainlikeimfive | Showerthoughts | NoStupidQuestions | Lightbulb |
|---|---|---|---|
| tipofmytongue | legaladvice | Advice | needadvice |
| answers | TooAfraidToAsk | AskReddit | askscience |
| AskCulinary | AskSocialScience | AskEngineers | TrueAskReddit |
| AskDocs | CrazyIdeas | explainlikeIAmA | AskHistorians |
| askphilosophy | ExplainLikeImCalvin | cscareerquestions | gamedev |
| engineering | biology | chemistry | Physics |
| FanTheories | gardening | writing | Survival |
| ReverseEngineering | Health | Cooking | slowcooking |
| socialskills | datascience | nutrition | |

Table 21: List of subreddits used for scraping