# OpenReview forum: "Reinforcement Replaces Supervision: Query focused Summarization using Deep Reinforcement Learning"
_EMNLP/2023/Conference — EMNLP 2023 Main_

### Official Review · Reviewer_ydP6 · 2023-08-04

**Soundness:** 4

**Excitement:**

4: Strong: This paper deepens the understanding of some phenomenon or lowers the barriers to an existing research direction.

**Paper Topic And Main Contributions:**

This paper focuses on Query-focused Summarization (QfS). This paper proposes RL-based models for QFS. In detail, this paper 1) uses Scheduled Sampling to resolve the conflict of employing RL in Transformers with Teacher Forcing, 2) presents a novel reward derived using the Cluster Hypothesis, 3) contributes the RPEDT dataset to train passage embedding models, and 4) contributes the RQFT dataset for analyses of QfS methods. The extensive experimental results show the effectiveness of the proposed methods.

**Reasons To Accept:**

1. A new RL-based method that solves the conflict of RL in Transformers with Teacher Forcing.
2. A new passage embedding-based reward mechanism using the Cluster Hypothesis.
3. Two new datasets for training and analysing models.

**Reasons To Reject:**

1. This paper only considers two baselines from 2019 and 2020. In a sense, these baselines may already be outdated and not keeping up with the current STOA techniques. I'm not an expert on QfS, but I think newer baselines in the recent 2 years should be considered.
2. As shown by Table 6, the authors do not report the experimental results of the two baselines on the new datasets created by the authors.


**Reproducibility:**

3: Could reproduce the results with some difficulty. The settings of parameters are underspecified or subjectively determined; the training/evaluation data are not widely available.

**Reviewer Confidence:**

3: Pretty sure, but there's a chance I missed something. Although I have a good feel for this area in general, I did not carefully check the paper's details, e.g., the math, experimental design, or novelty.

---

> ### Author Rebuttal · Authors · 2023-08-29
>
> We thank the reviewer for their time and their insightful comments and questions. We have attempted to address the questions to the best of our ability given the time constraints.
>
> > Q1. Recent baselines and SOTA techniques.
>
> -- A1. We present our results on two standard benchmarks -- ELI5 (**Table 6**) and DebatePedia (**Table 9**). DebatePedia has recent baselines, hence we report them and it can be seen we improve over the baselines with our method, in **zero shot setting too** (model trained on ELI5 train set and tested on DebatePedia) -- proving our approaches' superiority the over SOTA techniques for QfS. As for ELI5, recent papers use the KILT version of the dataset, which poses the query-summary pairs in an Information Retrieval format, with no gold support document to summarize from. We use the most recent available baselines. Please refer to **Lines 132-136** and **Lines 415-420** in the paper for justification.
>
> > Q2. Table 6 experimental results not shown for baselines.
>
> -- A2. The baselines do not release their trained models, which is why results on the new dataset could not be obtained. However, we retrain a baseline model (replicating the one by Lewis et al., 2020) -- BART SL, and produce the results in **Table 6**, specifically **Row 3**.

---

### Official Review · Reviewer_G1x8 · 2023-08-08

**Soundness:** 3

**Excitement:**

3: Ambivalent: It has merits (e.g., it reports state-of-the-art results, the idea is nice), but there are key weaknesses (e.g., it describes incremental work), and it can significantly benefit from another round of revision. However, I won't object to accepting it if my co-reviewers champion it.

**Missing References:**

- QMSum: A New Benchmark for Query-based Multi-domain Meeting Summarization (Zhong et al., 2021)
- SOCRATIC Pretraining: Question-Driven Pretraining for Controllable Summarization (Pagnoni et al., 2023)
- Interactive Query-Assisted Summarization via Deep Reinforcement Learning (Shapira et al, 2022)
- Exploring Neural Models for Query-Focused Summarization (Vig et al., 2022)
- Adapting pretrained text-to-text models for long text sequences. (Xiong et al., 2022)


**Paper Topic And Main Contributions:**

The paper is about Reinforcement Learning based models for the task of QFS. They use multiple rewards with Policy Gradient networks, using signals such as ROUGE, BLEU, and Semantic Similarity,

**Questions For The Authors:**

- Line 15: Be specific on which datasets.


**Reasons To Accept:**

- The curated datasets (RPEDT for passage embedding models and RQFT for analyses of QFS models) could be useful as a testbed for future research.

- The interesting discovery that RL aids in the acquisition of QFS by the model, even while considering the ROUGE score. Also, the Cluster Hypothesis outperforms rival passage embedders in providing enhanced semantic feedback.

**Reasons To Reject:**

- The most common QFS evaluation datasets are QMSum and SQuALITY. I would like to see the comparison w/ SL and w/ RL there, and compare to those existing results reported, for example, BART-LS, BART-Large SegEnc, and BART-Large SegEnc + SOCRATIC Pretraining.

- The authors might want to double-check whether the claim in line 84 is correct. For example, "Teacher Forcing Recovers Reward Functions for Text Generation (Hao et al., 2022)" and some of the papers in their related work might be a good starting point to check more details. I am not very familiar with that line of research.

**Reproducibility:**

3: Could reproduce the results with some difficulty. The settings of parameters are underspecified or subjectively determined; the training/evaluation data are not widely available.

**Reviewer Confidence:**

3: Pretty sure, but there's a chance I missed something. Although I have a good feel for this area in general, I did not carefully check the paper's details, e.g., the math, experimental design, or novelty.

---

> ### Author Rebuttal · Authors · 2023-08-29
>
> We thank the reviewer for their time and their insightful comments and questions. We have attempted to address the questions to the best of our ability given the time constraints.
>
> > Q1. QMSum, SQuALITY, and SOCRATIC Pretraining
>
> -- A1. We do not report results on QMSum and SQuALITY primarily for the following reason:
> * The document lengths, $9070$ and $5200$ respectively, exceed our models' maximum length limit ($1568$) by a huge margin. For reasons of time and space constraints of training our model, we select a dataset with decently sized documents ($1120$ tokens, average) and summaries ($150$ tokens, average).
>
> Additionally, the queries (questions) in the QMSum dataset come from a limited set of $13$ templates. In contrast to this, the queries in ELI5, DebatePedia, and our proposed test set are quite diverse. The significant improvements on these datasets thus validate that our proposed models generalize quite well.
>
> While we acknowledge the results of the recent (ACL 2023) paper on SOCRATIC pretraining, a fair comparison of RL-based methods for such long inputs would require significant changes to the approach (such as including the Segment Encoder, [[1]](https://aclanthology.org/2022.findings-naacl.109.pdf), to reduce resource requirement) -- indicating that `this is better addressed in a future work, rather than the presented work`.
>
> > Q2. The authors might want to double-check whether the claim in line 84 is correct. Check "Teacher Forcing Recovers Reward Functions for Text Generation (Hao et al., 2022)"
>
> -- A2. The cited work (and other works it cites) is totally orthogonal to our claim in line 84. Our claim is `we are the first work to resolve the conflict of employing RL in Transformers with Teacher Forcing`. Why is Teacher Forcing bad for application of RL to Natural Language Generation? $\rightarrow$ `Using ground truth tokens implies we are doing no exploration whatsoever, since we stay very close to the reference data set. The point of RL is to let the algorithm explore generations which are not exactly (token to token) the same as the ground truth, but are semantically similar.`
> The cited work discusses something orthogonal $\rightarrow$ It proves (theoretically) that a model trained using Teacher Forcing learns an Inverse Reinforcement Learning (IRL) model $\rightarrow$ basically a reward function -- `this in no way aids or contradicts our claim, nor does it address our claim`.
>
> > Q3. Missing References
>
> -- A3.
>
> * QMSum (Zhong et al., 2021) -- We omit this for reasons in A1.
> * Socratic Pretraining (Pagnoni et al., 2023) -- As addressed in A1, it is not a fair comparison with our proposed models. However, we acknowledge that this would work be a really strong comparison for future work, extending our approach to longer sequences. We thank the reviewer for this suggestion.
> * (Shapira et al, 2022) -- We omit this as it employs RL for **extractive** QfS, and we strictly focus on RL for **abstractive** QfS. However, thank you for the suggestion, and we will be including our literature survey on RL for extractive QfS in our camera-ready paper.
> * (Vig et al., 2022) -- Thank you for suggesting this, we will include it in the camera-ready paper.
> * (Xiong et al., 2022) -- This work presents a new LLM, and would have been related, had we included the QMSum benchmark. However, for the reasons stated above, we omit QMSum, which is why we do not include this work in our literature survey.
>
> ----
>
> * ### References
>
> [1] Exploring Neural Models for Query-Focused Summarization (Vig et al., 2022; In NAACL-2022)

---

### Official Review · Reviewer_VnCj · 2023-08-11

**Soundness:** 3

**Excitement:**

3: Ambivalent: It has merits (e.g., it reports state-of-the-art results, the idea is nice), but there are key weaknesses (e.g., it describes incremental work), and it can significantly benefit from another round of revision. However, I won't object to accepting it if my co-reviewers champion it.

**Paper Topic And Main Contributions:**

The paper "Reinforcement Replaces Supervision: Query focused Summarization using Deep Reinforcement Learning"  presents work on query focused summarization based on Reinforcement Learning, rather than Supervised Learning. In order to demonstrate the quality of the work, the paper presents a new dataset, human and automatic evaluation and the respective pipeline to achieve the results, which are also compared to two other datasets.

**Questions For The Authors:**

What is the rationale behind building the rewards on ROUGE and evaluating on ROUGE?

Is ROUGE an appropriate evaluation metric for these type of summaries?

Which implementation and version of ROUGE has been used and what parameters have been given?

Are the results and the reported improvements over other approaches statistically significant?

Are there any plans to publish the research artifacts for the research community?

**Reasons To Accept:**

The pipeline, the dataset and the comparison would be valuable to the community, if the research artifacts are release.

**Reasons To Reject:**

Building rewards based on ROUGE and evaluating based on ROUGE seems to be a kind of self-fulfilling prophecy.

**Reproducibility:**

1: Could not reproduce the results here no matter how hard they tried.

**Reviewer Confidence:**

4: Quite sure. I tried to check the important points carefully. It's unlikely, though conceivable, that I missed something that should affect my ratings.

**Typos Grammar Style And Presentation Improvements:**

l. 128 "inspite"
l. 226 "token-by-token match We take the RL..." missing full stop
l 520 "On the other hand...." -- First hand is missing

---

> ### Author Rebuttal · Authors · 2023-08-29
>
> We thank the reviewer for their time and their insightful comments and questions. We have attempted to address the questions to the best of our ability given the time constraints.
>
> > Q1. Building rewards on ROUGE and evaluating on ROUGE is a self-fulfilling prophecy -- rationale. Is ROUGE an appropriate evaluation metric?
>
> -- A1. Inconsistency between Train and Test measurement has been a long-standing challenge in Natural Language Generation $\rightarrow$ Abstractive Summarization ([[1]](https://openreview.net/pdf?id=HkAClQgA-)), Image Captioning ([[10]](https://arxiv.org/pdf/1612.00563.pdf)), Machine Translation ([[11]](https://openreview.net/pdf?id=HkG7hzyvf)), etc. The survey [[2]](https://arxiv.org/abs/1805.09461) covers this in more detail. All these works stress the requirement of consistency between the training and testing metrics.
> ROUGE ([[3]](https://aclanthology.org/W04-1013.pdf)) is a standard metric for evaluation of summaries ([[1]](https://openreview.net/pdf?id=HkAClQgA-), [[3]](https://aclanthology.org/W04-1013.pdf), [[4]](https://aclanthology.org/P17-1098.pdf), [[5]](https://aclanthology.org/K16-1028.pdf)), which highly correlates with the human judgement of the quality of summaries. Query focused Summarization also uses ROUGE as the de-facto metric for evaluation ([[4]](https://aclanthology.org/P17-1098.pdf), [[6]](https://aclanthology.org/2022.tacl-1.36/), [[7]](https://aclanthology.org/2021.naacl-main.393.pdf), [[8]](https://aclanthology.org/2022.findings-acl.61.pdf)). Hence, with the premises that `there should be consistency between train and test measurement`, and that `ROUGE is a proven True North metric for summarization (including query-focused summarization)`, optimizing on ROUGE is a logical conclusion. Thank you for pointing this out, we will include this justification in the camera-ready paper.
>
> > Q2. Which implementation and version of ROUGE has been used and what parameters have been given?
>
> -- A2. We use a ROUGE implementation in Python available publicly -- [https://pypi.org/project/rouge/](https://pypi.org/project/rouge/). We shortlist this particular implementation, as it has been used in the baselines ([[9]](https://aclanthology.org/P19-1346.pdf)), for a fair comparison. We keep the default parameters set by the library, similar to the baselines.
>
> > Q3. Are the results and the reported improvements over other approaches statistically significant?
>
> -- A3. Yes, the results are statistically significant. We report the statistical significance test results for the human evaluation metrics (Lines 491-496) -- which show that our proposed models provide statistically significant improvement over the baselines.
>
> > Q4. Are there any plans to publish the research artifacts for the research community?
>
> -- A4. Yes, we plan to release all artifacts, which include -- datasets, model checkpoints, and code.
>
> > Reproducibility Score 1
>
> -- Response: We report the architectures and all the hyperparameters used to train the model, along with the training infrastructure required -- please refer to sections G, H, I, and J in the Appendix for all the details. In our opinion, the entire project should be reproducible using the specified details.
>
> ----
>
> * ### References
>
> [1] A Deep Reinforced Model for Abstractive Summarization (Paulus et al., 2018; In ICLR-2018)
>
> [2] Deep Reinforcement Learning For Sequence to Sequence Models (Keneshloo et al., 2018)
>
> [3] ROUGE: A Package for Automatic Evaluation of Summaries (Lin, 2004; In ACL-2004)
>
> [4] Diversity driven attention model for query-based abstractive summarization (Nema et al., 2017; In ACL-2017)
>
> [5] Abstractive Text Summarization using Sequence-to-sequence RNNs and Beyond (Nallapati et al., 2016; In CoNLL-2016)
>
> [6] Document Summarization with Latent Queries (Xu and Lapata, 2021; In TACL-Volume 10, 2022)
>
> [7] Hurdles to Progress in Long-form Question Answering (Krishna et al., 2021; In NAACL-2021)
>
> [8] Read before Generate! Faithful Long Form Question Answering with Machine Reading (Su et al., 2022; In ACL-2022)
>
> [9] ELI5: Long Form Question Answering (Fan et al., 2019; In ACL-2019)
>
> [10] Self-critical Sequence Training for Image Captioning (Rennie et al., 2016; In CVPR-2017)
>
> [11] A differentiable BLEU loss. Analysis and first results (Casas et al., 2018; In ICLR-2018)

---

### Meta-Review · Area_Chair_M3JY · 2023-09-13

**Recommendation:** 4

**Metareview:**

The paper introduces Reinforcement Learning (RL) based models for Query-focused Summarization (QFS) inspired by RL in natural language generation. The proposed work employs rewards through ROUGE, BLEU, and Semantic Similarity. It addresses conflicts with Teacher Forcing through Scheduled Sampling. Additionally, the paper introduces a novel reward based on the Cluster Hypothesis, contributes the RPEDT dataset for passage embedding model training, and presents the RQFT dataset for QfS method analysis. The experimental results demonstrate the effectiveness of the proposed method in QFS. Please add baseline results on the new dataset for Fan et al. (2019).

---

### Decision · Program_Chairs · 2023-10-07

**Decision:**

Accept-Main

**Comment:**

The paper introduces Reinforcement Learning (RL) based models for Query-focused Summarization (QFS) inspired by RL in natural language generation. The proposed work employs rewards through ROUGE, BLEU, and Semantic Similarity. It addresses conflicts with Teacher Forcing through Scheduled Sampling. Additionally, the paper introduces a novel reward based on the Cluster Hypothesis, contributes the RPEDT dataset for passage embedding model training, and presents the RQFT dataset for QfS method analysis. The experimental results demonstrate the effectiveness of the proposed method in QFS. Please add baseline results on the new dataset for Fan et al. (2019).